# Multicentre Evaluation of Hepika Test Clinical Accuracy in Diagnosing HPV-Induced Cancer and Precancerous Lesions of the Uterine Cervix

**DOI:** 10.3390/diagnostics11040619

**Published:** 2021-03-30

**Authors:** Daniela Gustinucci, Lucia Ciccocioppo, Luigi Coppola, Giovanni Negri, Gianfranco Zannoni, Basilio Passamonti, Elena Cesarini, Ciro Ianzano, Tiziana Andreano, Anjuta Pireddu, Paolo Giorgi-Rossi

**Affiliations:** 1Laboratorio Unico di Screening USL Umbria 1, 06124 Perugia, Italy; bpassamonti1@gmail.com (B.P.); elena.cesarini@uslumbria1.it (E.C.); 2Cytopathology Unit, Renzetti Hospital, 66034 Lanciano, Italy; ciccocioppolucia@virgilio.it (L.C.); ciro.ianzano@gmail.com (C.I.); 3Pathology Unit, San Filippo Neri Hospital, 00135 Rome, Italy; luig.coppola@gmail.com; 4Pathology Unit, Central Hospital Bolzano, 39100 Bolzano, Italy; giovanni.negri@gmail.com; 5Department of Woman, Child and Public Health Sciences, Gynecopathology and Breast Pathology Unit, Catholic University of Sacred Hearth, 00168 Rome, Italy; gianfranco.zannoni@unicatt.it; 6Pathology Unit, Renzetti Hospital, 66034 Lanciano, Italy; tittola74@gmail.com; 7Pathology Unit, USL Umbria 1, 06012 Città di Castello, Italy; anjuta.pireddu@uslumbria1.it; 8Epidemiology Unit, Azienda Unità Sanitaria Locale—Istituto di Ricovero e Cura a Carattere Scientifico di Reggio Emilia, 42123 Reggio Emilia, Italy; paolo.giorgirossi@ausl.re.it

**Keywords:** Hepika, HPV, carcinoma, CIN, precancerous lesion, cancer, tumor biomarker, uterine cervix

## Abstract

Objective: To evaluate the clinical accuracy of Hepika test to identify cancer/precancerous lesions of the uterine cervix. Materials and Methods: A multicentre retrospective study was carried out in 2018 and included 330 liquid-based cytology samples from three Italian centres of women aged 25–64 who had been tested for the human papillomavirus (HPV) and whose histology or follow-up outcome was known. Hepika is an enzyme-linked immunosorbent assay (ELISA) targeting the protein complexes E6#p53 and E7#pRb. After excluding samples without sufficient residual material, the clinical accuracy of Hepika test was evaluated in 274 samples: adenocarcinoma (ADC) (4), squamous cell carcinoma (SCC) (7), adenocarcinoma in situ (AIS) (1), cervical intraepithelial neoplasia (CIN) grade 3 (60), CIN2 (51), CIN1 (34), and negative histology (117). Association, sensitivity, and specificity for carcinoma, CIN3+ and CIN2+ are reported. Results: Positive Hepika test was associated with a high probability of carcinoma (odds ratio (DOR) = 33.68, 95% confidence interval (CI) 7.0–163.1); sensitivity was 81.8%, specificity, 88.2%. A positive Hepika test showed a weaker association with CIN3+ lesions (DOR = 3.5; 95% CI 1.75–6.99) and lower sensitivity (27.8%). Conclusion: The Hepika test was found to be an accurate biomarker for HPV-induced cervical carcinoma. Population-based prospective studies are needed to confirm the clinical usefulness of the Hepika test in the differential diagnosis of HPV-induced invasive lesions.

## 1. Introduction

Uterine cervix carcinoma is the fourth most commonly diagnosed cancer in women worldwide, with about 570,000 new cases in 2018, causing 7.5% of deaths from cancer among women [1,2]. Squamous cell carcinoma (75%) and adenocarcinoma (10–15%) are the most frequent histotypes [3]. In high-income countries, the spread of the Pap test and implementation of screening programmes aimed at the early identification of precancerous lesions have led to a decrease in incidence and mortality rates [4,5,6].

According to results of trials showing better efficacy than the Pap test in reducing cervical cancer incidence (Ronco 2014), the human papillomavirus (HPV) test has been recommended by most international guidelines (US Preventive Services Task Force 2018; European Union (EU) guidelines 2015) and has become the routine primary screening test for cervical lesions in many high-income countries [7,8,9,10,11,12,13]. However, the problem of its low specificity persists, thus imposing the need for a triage test today [14,15].

In recent years, studies have been carried out to identify the exact mechanism of action of HPV in the genesis of uterine cervix carcinoma [16,17]. It has become clear that only cervical intraepithelial neoplasia grade 3 (CIN3) and, to a lesser degree, CIN2, may be cervical cancer precursors, with only a small part actually progressing to cancer [18]. Based on these findings, increasing attention has been paid to searching for a biomarker able to detect, in addition to the presence of viral DNA, molecular alterations produced by the virus itself on the cell. Being able to distinguish between those infections associated with, or which could lead in the short term to, an invasive carcinoma and those that could regress or not progress in the short term would radically transform screening programmes, thus reducing the number of treatments needed to prevent cancer and the related anxiety.

Viral proteins E6 and E7 are defined as oncoproteins since they interfere with the regulation of cell proliferation, promoting DNA instability [16,17,18,19] by acting on main regulation paths of the cell cycle [20]. The activity of these oncoproteins takes place through their link to two human oncosuppressor proteins (p53 and pRb) and represents a necessary step for cell transformation towards an invasive malignant tumour. Taking this into account, the protein complexes E6#p53 and E7#pRb may be ideal targets for a test capable of identifying invasive cancer lesions and those precancerous lesions at higher risk of progressing in the short term [21,22,23,24,25,26,27,28,29,30,31,32,33].

The immunoenzymatic test Hepika [*Hepika^®^ s.r.l. Avezzano (AQ) Italy*], a new biomarker, has been developed to search for the protein complexes E7#pRb and E6#p53. The Hepika test, related to the product of viral DNA expression, can be used as a specific biomarker for HPV-induced carcinoma by identifying the protein interactions E6#p53 and E7#pRb. The indication proposed for the use of this test is the diagnosis of invasive carcinoma to define surgical planning in cases of an undefined or a high-grade cytohistological diagnosis that does not exclude invasion.

The main objective of this study was to evaluate the accuracy of the Hepika test in diagnosing HPV-induced cancer (SCC, ADC, and AIS) and high-grade cervical intraepithelial neoplasia (CIN2 and CIN3) in cytology samples from diagnostic laboratories or organized cervical carcinoma screening programmes. 

The secondary objective was to evaluate whether test accuracy was affected by the cellularity of the sample.

## 2. Materials and Methods

The clinical accuracy of the Hepika test was evaluated by an Italian multicentre retrospective study carried out from 1 April to 30 June 2018. The Hepika test was performed on the unused residual cell odds of the 278 samples enrolled.

### 2.1. Study Population

Samples satisfying all the following inclusion criteria were considered suitable and analysed in the study: (1) samples from women between ages 25 and 65; (2) samples from diagnostic laboratories or organized cervical carcinoma screening programmes on which the Roche Cobas 4800 HR-HPV DNA (RHPV) test (Roche Diagnostics, GmbH, Mannheim, Germany) or the Qiagen HC2 HR-HPV DNA (QHPV) test (Qiagen Inc., Hilden, Germany) was performed; (3) liquid-based cervical cytology samples conserved in the following devices: ThinPrep^®^ PapTest (Hologic™, Marlborough, MA, USA), BD SurePath™ (BD Diagnostic, Burlington, NC, USA), or EasyPrep Gin (YD Diagnostic Corp, Yongin, Korea).

Exclusion criteria were as follows: (1) samples that, prior to the study, had not been conserved at a temperature between 2 °C and 30 °C from the sampling date; (2) samples not containing residual material with a noticeable sediment adequate for the setup of the Hepika test; (3) samples damaged during previous preparation procedures for molecular and/or cytological analysis.

The study was carried out on samples from diagnostic laboratories or organized screening programmes whose standard protocol [34] was based on the HPV test and triage cytology or on cytology and, for most positive samples (atypical squamous cells of undetermined significance [ASC-US] or low-grade squamous intraepithelial lesions [L-SIL]), on the HPV test. Samples were selected according to the eligibility criteria from case studies of the participating centres between 1 January 2013 and 31 March 2018. All HPV-positive samples meeting the inclusion criteria present at the moment the laboratories decided to take part in the study and all cytology-positive samples for which a triage HPV test had been performed were collected. This corresponds almost perfectly to samples of consecutive cytology-positive HPV cases in the Italian screening protocol, thus reducing the possibilities of selection bias and minimizing logistic issues concerning the shipping and conservation of the samples in comparison with a sample of consecutive cases. Moreover, 78 HPV DNA-negative consecutive samples collected from another study were included. Lastly, an HPV-negative case that turned out to be a squamous cell cancer during follow-up was included, although this case could not be considered a part of the sample of consecutive cases.

There were 330 eligible study samples, of which 82 (24.8%) were HPV-negative and 248 (75.2%) HPV-positive. In the pre-analytical phase, 52 samples were excluded (13 HPV-negative and 39 HPV-positive) since the residual material was inadequate for the setup of the Hepika test (Figure 1).

A total of 278 samples were included, of which 69 (25%) had a negative HPV test, 35 (13%) had a positive HPV test and negative cytology, and 174 (62%) had a positive HPV test and ASC-US+ cytology. Each centre participated with the sample odds agreed on with the coordinating centre, taking into account the inclusion criteria. Information (age, diagnosis, etc.) regarding the enrolled samples was previously acquired during the admission and diagnostic pathway of the women; information was registered in an anonymous form and sent to the coordinating centre.

The centres taking part in the study and providing the samples were the Laboratorio Unico di Screening of USL Umbria 1 of Perugia, Umbria Region, the Pathology Unit of San Filippo Neri Hospital of Rome, Lazio Region, and the Cytopathology Unit of Renzetti Hospital of Lanciano, Abruzzo Region. The Laboratorio Unico di Screening of Perugia acted as coordinator. Each participating centre applied different procedures to perform the Pap test and HPV test. All centres had already performed these tests on the samples, following the manufacturers’ operating instructions, with routine analytical procedures in their current version. The three centres had previously performed the following tests on the samples: the Laboratorio Unico di Screening: Roche Cobas 4800 HR-HPV DNA test and liquid-based cytology Hologic™ ThinPrep^®^ PapTest or Becton Dickinson SurePath™; the Pathology Unit of Rome: Qiagen HC2 HR-HPV DNA test and liquid-based cytology YD Diagnostic CORP EasyPrep Gin; the Cytopathology Unit of Lanciano: Qiagen HC2 HR-HPV DNA test and liquid-based cytology Hologic™ ThinPrep^®^ Pap test. All centres performed histological examination in the cases set out in the diagnostic protocol. The 2001 Bethesda classification [35] was used for the cytological diagnosis; the 2014 World Health Organization classification [36] was used for the histological diagnosis.

In the first 3 months of 2018, participating centres assigned the samples enrolled in the study a code (Centre ID) and a sequential number (Case No.) to anonymise them. All cases were identified by this code (Centre ID and Case No.), which was also used for labelling the sample. The centres applied the labels on biological samples before shipping them to the coordinating centre by courier.

The Laboratorio Unico di Screening of Perugia performed all the analyses with the Hepika test, following the kit operational instructions. As analysis procedures with the Hepika test require cell lysis for the extraction of proteins, it was not possible to preserve further sample odds. As this was a retrospective study, the results of the Hepika test were not provided to the woman or her primary physician since the woman had already completed her pathway according to the diagnostic-therapeutic protocol in place. All women gave their consent to use their samples for studies on the accuracy of biomarkers for cervical screening.

### 2.2. Definition of the Reference Standard

The reference endpoint was whichever histology was worse, colposcopy-guided biopsy or conization or hysterectomy, when performed. Cases were considered negative when colposcopy-guided biopsy was not performed because the colposcopy result was negative. Cases with a positive HPV test and negative cytology were followed up until a negative HPV test was achieved, and so considered negative, or to the confirmation of colposcopy-guided histology. Cases with low-grade cytology and a negative HPV test were also considered negative without colposcopy confirmation. Cases with only a negative HPV test were considered negative. No cases with high-grade cytology and a negative HPV were found during follow-up. The accuracy was calculated for different severity levels of the lesion: carcinoma, CIN3+, and CIN2+.

The evaluation of the new Hepika diagnostic test was carried out with histology as the standard diagnosis. As the final histology report represented a relevant factor for this study, specific measures for the quality check were established. An independent review board composed of two pathologists who were not aware of the molecular and immunoenzymatic test results, of the cytology and histology, or of the clinical conclusions verified the histological diagnoses. Participating centres sent the slides concerning biopsy, endocervical curettage, conization, or hysterectomy to the coordinating centre. The review process used slides stained originally with haematoxylin-eosin or tissue sections similar to the original ones on slides stained with immunohistochemistry for protein p16. The review of each histological diagnosis was assigned to the two board members in charge of delivering a morphological diagnosis on the preparation stained with haematoxylin-eosin, an immunohistochemistry evaluation for p16, and a final comprehensive diagnosis. When the final diagnosis of the review board agreed with the original diagnosis, the diagnosis was considered the final one for the study. In cases of discrepancy between the original diagnosis and the diagnosis provided by the two reviewers, the majority diagnosis was considered. Discordant cases between the original diagnosis and the diagnosis of the individual reviewers were assigned to a third review pathologist, who made the final diagnosis (Table 1).

### 2.3. Evaluation of Sample Adequacy and Cytology

After the exclusion of the samples without noticeable sediment, the evaluation of the quality of the residual cytological material was performed on all 109 ThinPrep samples with satisfactory cellularity in the slide and for which the residual liquid was at least 10 mL or at least 6 mL if two slides had been prepared from the same sample before inclusion in the study (Table 2). 

The 109 Pap tests were then re-examined to perform the count of cellularity in the slide and to obtain a parameter for the comparison with the results of the Hepika tests. The count procedure performed the quantization of total cellularity and atypical cells through the selection of 20 optical fields with 20× magnification. Cells present in a slide derived from the average use of about 5 mL of a liquid-based sample that was sampled with adequate cellularity. In general, the reproducibility of cytological preparation with ThinPrep Pap test method under these conditions allowed for a uniform replication on later cytological preparations [37]. For the estimation of cellularity, proportioning the total cell count and the atypical cells with the quantity of liquid, it is possible to assume the quantity of suspension cells in the residual liquid of samples. Therefore, the quantity of total and atypical cells was estimated as follows:
(a)Total number of *optical fields 20×* (FOV20×) in a slide = µm^2^ 314,000,000 of circular area of slide containing the cells/µm^2^ 350,000 of area of FOV20× = 897.(b1)Number of *total cells* estimated for FOV20× = Mean of the count of total cells found in 20 FOV20× of slide.(b2)Number of *atypical cells* estimated for FOV20× = Mean of the count of atypical cells found in 20 FOV20× of slide.(c1)*Total cells* present in a slide = (*a × b1*) = (*897 × b1*).(c2)*Atypical cells* present in a slide = (*a × b2*) = (*897 × b2*).(d1)*Total cells* present in residual sample = (*c1/5 mL × n mL* of residual liquid).(d2)*Atypical cells* present in residual sample = (*c2/5 mL × n mL* of residual liquid).

### 2.4. Hepika^®^ Test (CE-IVD)

Hepika is a diagnostic test manufactured to identify lesions induced by HPV infection with a high probability of being or becoming invasive. It is an indirect test based on enzyme-linked immunosorbent assay (ELISA) technology for simultaneous qualification of the protein complexes E6#p53 and E7#pRb, pathognomonic for ongoing carcinogenesis in HPV-induced lesions, from cell extracts of liquid-based cytology samples [21,22]. Cytology samples must be conserved in approved methanol- or ethanol-based devices for liquid-based cytology, keeping the target proteins [e.g., ThinPrep^®^ Pap test (Hologic™), SurePath™ (Becton Dickinson), or similar] unaltered, following the specific manufacturer’s instructions also for executing the sampling.

ELISA technology requires the sensitization of wells of the plates with monoclonal antibodies specific against E6 and E7; here, previously lysed cell samples are incubated. During incubation, the complexes E6#p53 and E7#pRb are kept by monoclonal antibodies and, in subsequent phases, recognized by polyclonal antibodies. Immunocomplexes are recognized thanks to an HRP-conjugate antibody mixture. The detection phase is carried out by incubating the plate after having dispensed the chromogen ABTS [2.2’-azino-bis(3-etilbenzthiazoline-6-solfonic acid)] and by reading the optical density values with ELISA microplate reader. Positive checks are made of lyophilized extracts of HeLa cells [30,31]. Negative checks are made of lyophilized extracts of cells not expressing the targets. The kit sets out a single plate intended for the simultaneous detection of both protein complexes (E6#p53 and E7#pRb) for the analysis of 22 biological samples for each plate. Checks define the admittance criteria for the single test. Final results are interpreted by calculating the mean of optical densities detected in two wells of the sample and by comparing it with cutoffs: E6#p53: positive (mean) ≥ 0.160; E7#pRb: positive (mean) ≥ 0.160. A negative result of Hepika test indicates that one or both protein complexes (E6#p53 and/or E7#pRb) is/are absent; a positive result indicates that both protein complexes E6#p53 and E7#pRb are present.

### 2.5. Human Papillomavirus (HPV) DNA Test

HPV DNA Roche Cobas 4800 [38] and Qiagen Digene *HC2* [39] tests were performed according to the manufacturers’ instructions, as were the conservation and transport of the sample, following the analytical procedures per routine in their current version.

### 2.6. Statistical Analysis

MedCalc statistical software was used to perform the statistical analysis [https://www.medcalc.org/calc/diagnostic_test.php] (accessed on 30 January 2019). The chi-square test was used to evaluate the variability of categories [https://www.socscistatistics.com/tests/] (accessed on 30 January 2019). The Mann–Whitney U test was used to compare the means of parametric counts between the groups to evaluate the quality of the sample [https://www.socscistatistics.com/tests/] (accessed on 30 January 2019). The distribution of the parametric variables is represented by the median and interquartile range (P_1/4_–P_3/4_). Sensitivity, specificity, and positive and negative predictive values are presented as percentages, with 95% confidence intervals, calculated based on the binomial exact distribution. The diagnostic odds ratio (DOR) is also reported, which was calculated as the ratio between positive likelihood ratio divided by negative likelihood ratio of the test. The associated *p*-value was the probability of observing such an OR equal or even more distant from 1 with a sample of the same size.

## 3. Results

The Hepika test was positive in 40 cases (14.4%)-37 HPV-positive (13.3%) and 3 HPV-negative (1.1%)-with the following cytological diagnoses: ADC 100% (2/2), SCC 40% (2/5), high-grade squamous intraepithelial lesion (HSIL) 16.3% (15/92), low-grade squamous intraepithelial lesion (LSIL) 12.5% (8/64), atypical squamous cells-cannot exclude HSIL (ASC-H) 20% (2/10), atypical glandular cells (AGC) 50% (1/2), and negative cytology 20% (7/35) (Table 3).

The diagnostic investigation made it possible to define the disease state in 274 cases, with the following definitive histological diagnosis established by the review board: ADC 4 (1.5%), SCC 7 (2.6%), AIS 1 (0.4%), high-grade squamous intraepithelial lesions (CIN2/3) 111 (40.5%), low-grade squamous intraepithelial lesions (CIN1) 34 (12.4%), and negative histology 117 (42.7%) (Table 1).

A total of 14.6% of the samples were Hepika-positive with the following histological diagnoses: ADC 4 (100%), SCC 5 (71.4%), CIN3–AIS 11 (18%), CIN2 8 (15.7%), CIN1 7 (26.6%), and negative histology 5 (4.3%). All Hepika-positive samples were also HPV-positive (13.5%), except for three HPV-negative cases (1.1%), in one of which the follow-up with hysterectomy showed an invasive SCC (Table 4).

Accuracy parameters of the Hepika test, with 95% confidence intervals, were calculated for diagnoses of carcinoma, CIN3+, and CIN2+ (Table 5).

For carcinoma, we observed 81.8% sensitivity, and 88.2% specificity, the corresponding NPV was 99.1%, and positive likelihood ratio was 6.94. Considering all high-grade lesions, Hepika sensitivity was lower (CIN3+ 27.8%; CIN2+ 22.8%), with high specificity (CIN3+ 90.1%; CIN2+ 92.1%) and positive predictive values of 50% and 70% for CIN3+ and CIN2+, respectively. The high sensitivity and specificity of the Hepika test for carcinoma was also confirmed by the high probability (DOR = 33.68, *p* < 0.001) of this diagnosis being associated with a Hepika-positive test (Table 6).

### Adequacy of the Sample

Parameters of the subsample of 109 samples of the study reported in Table 2 were analysed. On average, the residual liquid was 11.3 mL, with mean values varying between 11.0 mL in CIN3 and 12.7 mL in carcinomas (Figure 2).

On average, total cellularity was 112,845 units, with mean values varying between 103,331 in CIN3 and 137,532 in CIN1 (Figure 3).

The number of atypical cells observed in cytology varied between 1184 (0.9%) and 173,463 (79.0%), with a mean of 21,568 (21.2%), distributed with decreasing mean values from carcinomas (41,859) to CIN1 (12,606), with a 2.3:1 carcinoma/CIN ratio. Their distribution (Figure 4) in single diagnostic categories increased with the severity of the lesion; higher median values were seen in Hepika-positive cases.

The mean quantity of atypical cells showed statistically significant differences (*p* < 0.05) in Hepika-positive vs. Hepika-negative groups, with divergent median values of 27,318 vs. 10,546 (Figure 5).

Figure 6 report the distribution of optical density values for samples below and over the Hepika positivity threshold, while Appendix A reports the optical density distribution by histological endpoint and shows a markedly different distribution for carcinoma.

## 4. Discussion

The Hepika test showed good sensitivity and specificity for invasive carcinoma, while its sensitivity for CIN3 and CIN2 was lower than that obtained by other biomarkers eligible for the HPV triage, such as cytology and p16/Ki67 dual staining (about 25% for Hepika and about 60–80% for the other biomarkers) [40,41,42,43,44,45,46,47]. A commercially available test targeting specifically the E6 protein [48] showed much higher sensitivity for CIN3 than Hepika, but it also showed 50% positivity in HPV-positive/histology negative and 16% positivity in HPV negative samples. Such a low sensitivity excludes a possible use of Hepika as a triage test for HPV-positive women, because HPV-positive/Hepika-negative would require too strict a follow up given the high prevalence of CIN2 and CIN3 in this group.

Hepika had a low sensitivity for precancerous lesions, which implies that in many of these lesions it was not possible to detect complexes E6#p53 and E7#pRb in atypical cells. These cells are likely to be in the initial phase of carcinogenesis. Indeed, progressive alterations of the cell cycle arising from oncoproteins E6 and E7, transcribed by integrated viral DNA, develop through the interaction with the oncosuppressor proteins p53 and pRb, causing the atypical morphological modifications of the cell: dysplasia or precancerous lesion in the initial phase of carcinogenesis and preinvasive or invasive neoplasia in an advanced phase of carcinogenesis [49,50,51]. In the initial phase of carcinogenesis, oncogenic progression starts with the expression of at least one of the aforementioned protein complexes [16,17,22], giving rise to dysplastic atypical cells. However, not all these lesions progress to cancer, and alterations of the cell cycle in this phase are still reversible. Therefore, the presence of morphologically atypical cells demonstrates the existence of pathological tissue that is neoplastic, which to transform into cancer needs the persistence of viral DNA that codifies the oncoproteins E6 and E7 and the relevant continuous expression of both protein complexes E6#p53 and E7#pRb [16,17,22,23,24,52,53,54]. Hepika, therefore, would identify the “atypical” keratinocyte that, in addition to having viral oncogenes, has been subjected to transformation into a “malignant tumour cell” expressing the protein complexes. The high E6#p53 and E7#pRb positivity of invasive lesions could suggest that histologically preinvasive lesions that are positive for these complexes, even if histologically classifiable as dysplasia, have already acquired molecular characteristics that are typical of invasive disease. Interestingly, an HPV-negative case that was diagnosed as SCC during follow-up was included in the sample and was Hepika-positive. HPV DNA negativity can be the consequence of analytic issues, of deletions in the viral DNA that occurred during integration into the host genome, or the lesion can be induced by a viral strain with an uncertain oncogenic potential not included in HPV DNA tests.

Hepika was positive in three out of 68 high-risk HPV-negative samples (4.4%) for which further investigations were not performed. This positivity in HPV-negative samples had a negative impact on specificity, even if false positives observed in a low risk population are not a real issue for a test that is a candidate for guiding treatment and not for screening. 

All analyses carried out to guarantee the adequacy of samples were satisfactory, with an adequate quantity of atypical cells that did not affect the accuracy of the study.

## 5. Conclusions

The high sensitivity for invasive carcinoma and the low sensitivity for CIN3 and CIN2 make Hepika a good candidate for identifying those lesions that can be selected for follow-up instead of immediate treatment in women who want to delay the treatment. In contrast, in a situation in which a high-grade lesion is suspected but for which histological confirmation is not possible, Hepika positivity could guide the decision on treatment. Lastly, in conization planning, in the presence of cytology and/or histology not excluding or not confirming suspected invasiveness, Hepika positivity could support a more radical approach.

Taking into consideration the limits of this retrospective study, other prospective studies are needed to confirm the diagnostic accuracy of the Hepika test. Moreover, in light of its application in the context of precancerous lesions, this new diagnostic test lends itself to further trials aimed at a strict surveillance over time of women with a negative Hepika test with untreated CIN2/3 to confirm whether this negativity may predict regression, and to follow up of women with CIN1 and a positive Hepika test to evaluate the probability of progression towards CIN3 of these lesions.

## Figures and Tables

**Figure 1 diagnostics-11-00619-f001:**
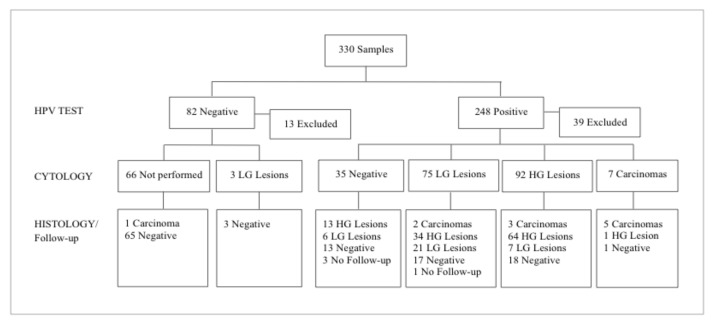
Samples of the study and relative testing results. (Abbreviations: HG: high grade; LG: low grade).

**Figure 2 diagnostics-11-00619-f002:**
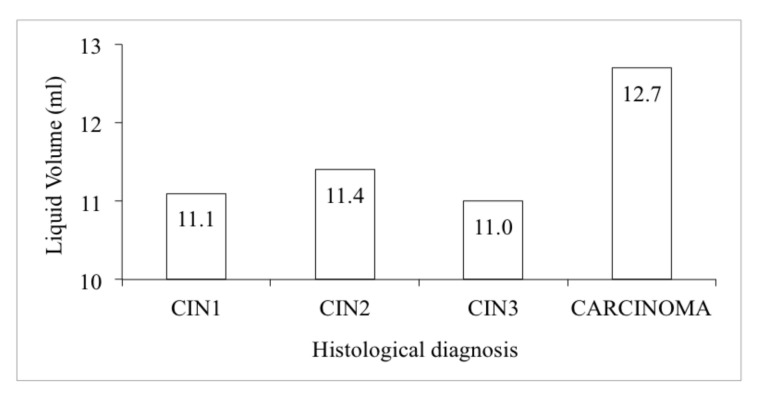
Mean size of residual liquid by histology.

**Figure 3 diagnostics-11-00619-f003:**
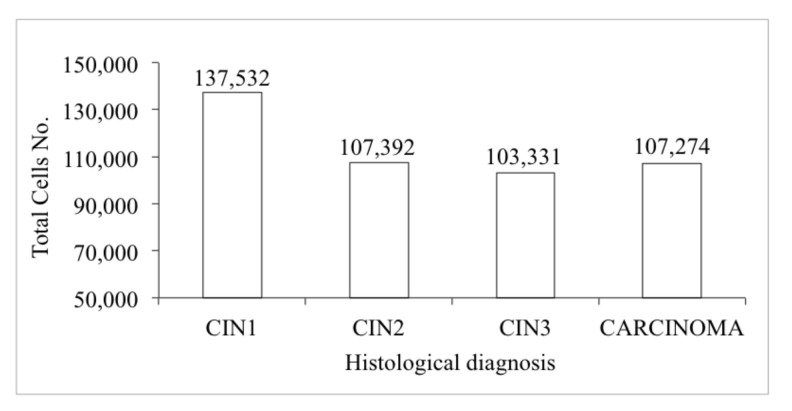
Estimated mean values of total cells by histology.

**Figure 4 diagnostics-11-00619-f004:**
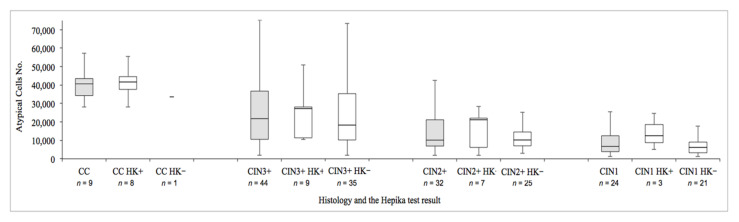
Distribution of atypical cells based on histology and Hepika test result. Data are presented in boxes and whiskers’ style, which represents the medians and ranges of the data. (Abbreviations: CC: carcinoma; CIN: cervical intraepithelial neoplasia (grade 1, 2, and 3); HK: Hepika (positive: +; negative: −); n: number of cases).

**Figure 5 diagnostics-11-00619-f005:**
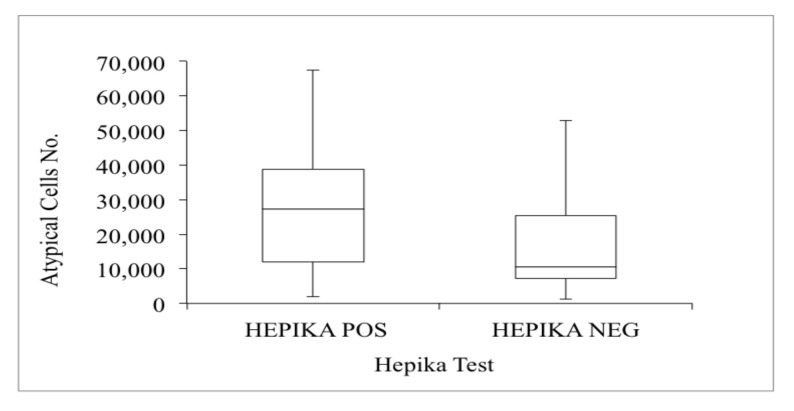
Distribution of atypical cells by Hepika test result. Data are presented in boxes and whiskers’ style, which represents the medians and ranges of the data. (Abbreviations: POS: positive; NEG: negative).

**Figure 6 diagnostics-11-00619-f006:**
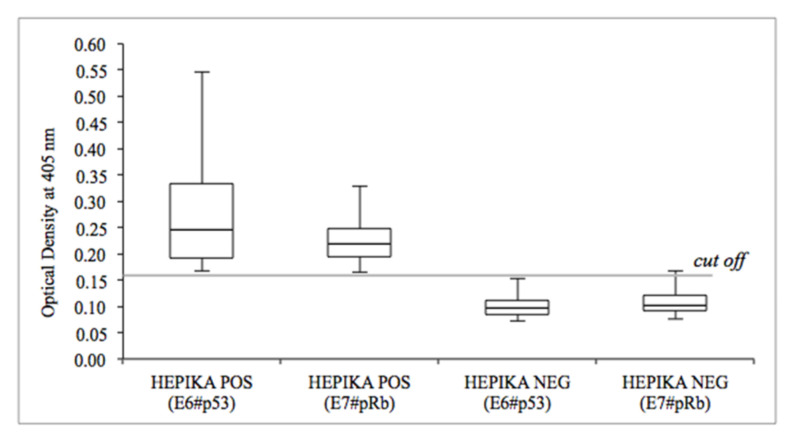
Distribution of complexes E6#p53 and E7#pRb optical density in relation to Hepika test result. Data are presented in boxes and whiskers’ style, which represents the medians and ranges of the data. (Abbreviations: POS: positive; NEG: negative).

**Table 1 diagnostics-11-00619-t001:** Correlation of molecular, cytological, and histological diagnoses of the samples selected for the study.

*Histology*	*HPV*				*Cytology*									*Total*	*%*
	*POS*	*%*	*NEG*	*%*	*ADC*	*SCC*	*HSIL*	*LSIL*	*ASC-H*	*ASC-US*	*AGC*	*NEG*	*N.P.*		
ADC	4	1.5%			2				1		1			4	1.5%
SCC	6	2.2%	1	0.4%		3	3						1	7	2.6%
AIS	1	0.4%					1							1	0.4%
CIN3	60	21.9%				1	43	9	2			5		60	21.9%
CIN2	51	18.6%					20	19	4			8		51	18.6%
CIN1	34	12.4%					7	19	2			6		34	12.4%
NEG	49	17.9%	68	24.8%		1	18	16	1	2	1	13	65	117	42.7%
Total	205	74.8%	69	25.2%	2	5	92	63	10	2	2	32	66	274	100.0%
					0.7%	1.8%	33.6%	23.0%	3.6%	0.7%	0.7%	11.7%	24.1%	100.0%	

Abbreviations: ADC: adenocarcinoma; SCC: squamous cell carcinoma; AIS: adenocarcinoma in situ; CIN3: squamous intraepithelial lesion grade 3; CIN2: squamous intraepithelial lesion grade 2; CIN1: squamous intraepithelial lesion grade 1; HSIL: high-grade squamous intraepithelial lesion; LSIL: low-grade squamous intraepithelial lesion; ASC-H: atypical squamous cells–cannot exclude HSIL; ASC-US: atypical squamous cells of undetermined significance; AGC: atypical glandular cells; POS: positive; NEG: negative; N.P.: not performed.

**Table 2 diagnostics-11-00619-t002:** Mean values of detections in 109 samples based on Hepika test and histology.

*HEPIKA Test*	*No.* *Cases*	*Histology*	*Residual Liquid* *Volume (mL)*	*Total Cells* *No. Estimation*	*Atypical Cells No.* *Estimation (%)*	*Optical Density* *E6#p53*	*Optical Density* *E7#pRb*
POSITIVE (27 Total)	4	ADC	13	151,287	47,300 (43%)	0.2985	0.2534
	4	SCC	12	78,271	38,505 (58%)	0.4731	0.3297
	9	CIN3	13	142,661	24,224 (17%)	0.2547	0.1993
	7	CIN2	13	140,494	15,473 (12%)	0.2190	0.2136
	3	CIN1	8	138,842	14,055 (17%)	0.1878	0.2133
		*Total*	*12*	*133,413*	*26,360 (26%)*	*0.2769*	*0.2319*
NEGATIVE (82 Total)	1	SCC	15	47,235	33,508 (71%)	0.1308	0.1430
	1	AIS	15	219,621	173,463 (79%)	0.2456	0.0967
	34	CIN3	10	89,499	22,643 (26%)	0.1015	0.1127
	25	CIN2	11	98,123	16,122 (15%)	0.1018	0.1099
	21	CIN1	11	140,874	12,399 (9%)	0.1027	0.1104
		Total	11	106,357	20,003 (19%)	0.1040	0.1114

Abbreviations: ADC: adenocarcinoma; SCC: squamous cell carcinoma; AIS: adenocarcinoma in situ; CIN3: squamous intraepithelial lesion grade 3; CIN2: squamous intraepithelial lesion grade 2; CIN1: squamous intraepithelial lesion grade 1.

**Table 3 diagnostics-11-00619-t003:** Hepika test results per cytology and human papillomavirus (HPV) DNA test.

*Cytology*	*No. Cases*	*Hepika*				*HK+*				*HK−*			
		*POS*	*%*	*NEG*	*%*	*HPV+*	*%*	*HPV−*	*%*	*HPV+*	*%*	*HPV−*	*%*
ADC	0	2	0.7%			2	100.0%						
SCC	2	2	0.7%	3	1.1%	2	40.0%			3	60.0%		
HSIL	5	15	5.4%	77	27.7%	15	16.3%			77	83.7%		
LSIL	92	8	2.9%	56	20.1%	8	12.5%			53	82.8%	3	4.7%
ASC-H	64	2	0.7%	8	2.9%	2	20.0%			8	80.0%		
ASC-US	10			2	0.7%					2	100.0%		
AGC	2	1	0.4%	1	0.4%	1	50.0%			1	50.0%		
NEG	35	7	2.5%	28	10.1%	7	20.0%			28	80.0%		
N.P.	66	3	1.1%	63	22.7%			3	4.5%			63	95.5%
Total	210	37	14.4%	175	85.6%	37	13.3%	3	1.1%	172	61.9%	3	23.7%

Abbreviations: HK+: Hepika-positive; HK−: Hepika-negative; HPV+: HPV-positive; HPV−: HPV-negative; ADC: adenocarcinoma; SCC: squamous cell carcinoma; HSIL: high-grade squamous intraepithelial lesion; LSIL: low-grade squamous intraepithelial lesion; ASC-H: atypical squamous cells—cannot exclude HSIL; ASC-US: atypical squamous cells of undetermined significance; AGC: atypical glandular cells; POS: positive; NEG: negative; N.P.: not performed.

**Table 4 diagnostics-11-00619-t004:** Hepika test results based on HPV test and review histology.

*Histology*	*No. Cases*	%	*Hepika*				*HK+*				*HK−*			
			*POS*	*%*	*NEG*	*%*	*HPV+*	*%*	*HPV−*	*%*	*HPV+*	*%*	*HPV−*	*%*
ADC	0	1.5%	4	100.0%			4	100.0%						
SCC	4	2.6%	5	71.4%	2	28.6%	4	57.1%	1	14.3%	2	28.6%		
CIN3–AIS	61	22.3%	11	18.0%	50	82.0%	11	18.0%			50	82.0%		
CIN2	51	18.6%	8	15.7%	43	84.3%	8	15.7%			43	84.3%		
CIN1	34	12.4%	7	26.6%	27	79.4%	7	20.6%			27	79.4%		
NEG	117	42.7%	5	4.3%	112	95.7%	3	2.6%	2	1.7%	46	39.3%	66	56.4%
Total	274	100.0%	40	14.6%	234	85.4%	37	13.5%	3	1.1%	168	61.3%	66	24.1%

Abbreviations: HK+: Hepika-positive; HK−: Hepika-negative; HPV+: HPV-positive; HPV−: HPV-negative; ADC: adenocarcinoma; SCC: squamous cell carcinoma; AIS: adenocarcinoma in situ; CIN3: squamous intraepithelial lesion grade 3; CIN2: squamous intraepithelial lesion grade 2; CIN1: squamous intraepithelial lesion grade 1; NEG: negative.

**Table 5 diagnostics-11-00619-t005:** Hepika test accuracy in function of histology (95% confidence interval (CI)).

*Histology*	*Sensitivity* *[% (95% CI)]*	*Specificity* *[% (95% CI)]*	*PPV* *[% (95% CI)]*	*NPV* *[% (95% CI)]*	*LR+* *[% (95% CI)]*	*LR−* *[% (95% CI)]*	*Prevalence*
CARCINOMA	**81.8%** (48%–98%)	**88.2%** (84%–92%)	**22.5%** (16%–31%)	**99.1%** (97%–100%)	**6.94** (4.50–10.70)	**0.21** (0.06–0.72)	**4.0%**
CIN3+	**27.8%** (18%–40%)	**90.1%** (85%–94%)	**50.0%** (36%–64%)	**77.8%** (75%–80%)	**2.81** (1.61–4.90)	**0.80** (0.69–0.93)	**26.3%**
CIN2+	**22.8%** (16%–31%)	**92.1%** (86%–96%)	**70.0%** (55%–81%)	**59.4%** (57%–62%)	**2.86** (1.52–5.39)	**0.84** (0.75–0.93)	**44.9%**

Abbreviations: CIN3+: squamous intraepithelial lesion grade 3 or worse; CIN2+: squamous intraepithelial lesion grade 2 or worse; PPV: positive predictive value; NPV: negative predictive value; LR+: positive likelihood ratio; LR-: negative likelihood ratio.

**Table 6 diagnostics-11-00619-t006:** Relationship between Hepika test and severity grade of cervical lesions.

	*Hepika POS*		*DOR*	*p Value*
*Histology*	*No. Cases*	*%*	*(95% CI)*	
CARCINOMA	**9**	**81.8** (9/11)	**33.68** (6.95–163.07)	*p < 0.001*
CIN3+	**20**	**27.8** (20/72)	**3.50** (1.75–6.99)	*p < 0.001*
CIN2+	**28**	**22.8** (28/123)	**3.41** (1.65–7.05)	*p < 0.001*

Abbreviations: CIN3+: squamous intraepithelial lesion grade 3 or worse; CIN2+: squamous intraepithelial lesion grade 2 or worse; POS: positive; NEG: negative; DOR: diagnostic odds ratio.

## Data Availability

The data, central to support in this study, are available in the body of the manuscript and Tables and Figures.

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
