# Peer review of "Multicentre Evaluation of Hepika Test Clinical Accuracy in Diagnosing HPV-Induced Cancer and Precancerous Lesions of the Uterine Cervix"

_diagnostics, 2021, doi:10.3390/diagnostics11040619_

Round 1

Reviewer 1 Report

In this manuscript, the authors evaluate the clinical accuracy of Hepika test to identify cancer/precancerous 16 lesions of the uterine cervix. The results are clear and manuscript is well documented. Some points should be revised before publication.

  1. The authors should discuss the difference between Hepika test and E6/E7 ELISA and why Hepika test is better?
  2. Graphic abstract should be provided for easier understand for readers.

Author Response

Reviewer # 1

The authors should discuss the difference between Hepika test and E6/E7 ELISA and why Hepika test is better?

RE: we are not presenting data useful for a head to head comparison of Hepika with any other test. We can only compare the rationale of the Hepika test with that of other test to explain  its possible clinical utility. The presented data confirm that Hepika is promising for identify lesions that are very likely to progress or  are already cancer.

We added two sentences in the discussion about the different possible use compared with other available tests. In the first sentence we also explained the difference between Hepika and the commercially available ELISA test targeting E6. (Yang YS, Smith-McCune K, Darragh TM, Lai Y, Lin JH, Chang TC, Guo HY, Kesler T, Carter A, Castle PE, Cheng S. Direct human papillomavirus E6 whole-cell enzyme-linked immunosorbent assay for objective measurement of E6 oncoproteins in cytology samples. Clin Vaccine Immunol. 2012 Sep;19(9):1474-9. doi: 10.1128/CVI.00388-12. Epub 2012 Jul 18. PMID: 22815148; PMCID: PMC3428408):

“The Hepika test showed good sensitivity and specificity for invasive carcinoma, while its sensitivity for CIN3 and CIN2 was lower than that obtained by other biomarkers eligible for the HPV triage, such as cytology and p16/Ki67 dual staining (about 25% for Hepika and about 60%–80% for the other biomarkers) [40-47]. Such a low sensitivity exclude a possible use of Hepika as triage test for HPV-positive women, because HPV-positive/Hepika-negative would require too strict follow up given the high prevalence of CIN2 and CIN3 in this group.”

And

“Hepika was positive in three out of 68 high-risk HPV-negative samples (4.4%) for which further investigations were not performed. This positivity in HPV-negative samples had a negative impact on specificity, even if false positive observed in a low risk population are not a real issue for a test that is candidate for guiding treatment and not for screening.”

2) Graphic abstract should be provided for easier understand for readers.

RE: we propose a simplified version of table 5 as graphical abstract.

Reviewer 2 Report

The authors investigated the Hepika test to detect cancer lesions of the uterine cervix. The study design is solid and the manuscript is well prepared. However, I have some points for the authors to consider.

- I would suggest the authors to analysis more analytical performance variables by biostatistical applications like Cohen kappa, diagnostic odds ratio, test scoring and clinical utility, precision, accuracy and so on. They consider as a table or adding to current tables.

- It is suggested to update tables by showing and comparing the results focus on approved methods with Hepika Test, for better understanding.

Author Response

Reviewer #2

1) I would suggest the authors to analysis more analytical performance variables by biostatistical applications like Cohen kappa, diagnostic odds ratio, test scoring and clinical utility, precision, accuracy and so on. They consider as a table or adding to current tables.

RE: in this study we do not report data on reproducibility neither an head to head comparison of two tests with the same use in clinical practice. Therefore we cannot use Cohen kappa to measure the inter-laboratory reproducibility of Hepika neither to measure the concordance between Hepica and another test targeting the same molecular change. We report the diagnostic odds ratio in table 6, even it was not called DOR. We now adopted this naming, that is actually more standardized, and we thank the reviewer for this suggestion.

All the measures of accuracy (sensitivity, specificity, negative and positive predictive value), likelihood ratio in positives and in negatives are reported in table 5.

The Hepika assay is presented here with a pre-defined positivity threshold, targeting binary outcomes, precision measures are not the scope of this paper.

Regarding clinical utility, we added a sentence in the discussion explain how the test could be used to change women management. In the conclusion we already mentioned which kind of study is needed to assess the potential clinical utility of Hepika.

2) It is suggested to update tables by showing and comparing the results focus on approved methods with Hepika Test, for better understanding.

RE: In this study we did not compare Hepika with other approved tests. Furthermore, the characteristics of Hepika, i.e. being very specific and sensitive for carcinoma but not sensitive for CIN2 and CIN3, suggest a possible clinical use in guiding surgical planning and in discouraging or not surveillance of CIN2, a role that has not been proposed for any other biomarker on the market. In fact all others biomarkers have been proposed for primary screening or for triaging HPV positive women.